# piRNAs and Aubergine cooperate with Wispy poly(A) polymerase to stabilize mRNAs in the germ plasm

Jérémy Dufourt[1], Gwénaëlle Bontonou[1], Aymeric Chartier[1], Camille Jahan[1], Anne-Cécile Meunier[1], Stéphanie Pierson[1], Paul F. Harrison [2,3], Catherine Papin[1], Traude H. Beilharz[3] & Martine Simonelig [1]

Piwi-interacting RNAs (piRNAs) and PIWI proteins play a crucial role in germ cells by repressing transposable elements and regulating gene expression. In *Drosophila*, maternal piRNAs are loaded into the embryo mostly bound to the PIWI protein Aubergine (Aub). Aub targets maternal mRNAs through incomplete base-pairing with piRNAs and can induce their destabilization in the somatic part of the embryo. Paradoxically, these Aub-dependent unstable mRNAs encode germ cell determinants that are selectively stabilized in the germ plasm. Here we show that piRNAs and Aub actively protect germ cell mRNAs in the germ plasm. Aub directly interacts with the germline-specific poly(A) polymerase Wispy, thus leading to mRNA polyadenylation and stabilization in the germ plasm. These results reveal a role for piRNAs in mRNA stabilization and identify Aub as an interactor of Wispy for mRNA polyadenylation. They further highlight the role of Aub and piRNAs in embryonic patterning through two opposite functions.

[1] mRNA Regulation and Development, Institute of Human Genetics, UMR9002 CNRS-University of Montpellier, 141 rue de la Cardonille, 34396 Montpellier Cedex 5, France. [2] Monash Bioinformatics Platform, Monash University, Melbourne, VIC 3800, Australia. [3] Development and Stem Cells Program, Monash Biomedicine Discovery Institute, Monash University, Melbourne, VIC 3800, Australia. Catherine Papin Deceased. Correspondence and requests for materials should be addressed to M.S. (email: Martine.Simonelig@igh.cnrs.fr)

Germ granules are specific ribonucleoprotein granules found in germ cells of all species. They contain mRNAs that have essential functions in germ cell specification and/or development[1]. In *Drosophila*, the germ plasm starts to assemble during mid-oogenesis with the localization and translation, at the posterior pole of the oocyte, of *oskar* (*osk*) mRNA that encodes the primary determinant of the germ plasm. Other maternal mRNAs then localize to the germ plasm using different mechanisms[2]. First, mRNA localization involves a diffusion and anchoring mechanism taking place at late stages of oogenesis, during nurse cell dumping, when nurse cells empty their content into the oocyte[3,4]. This mechanism is very inefficient, resulting in germ plasm localization of only 4% of maternal *nanos* (*nos*) mRNA, which encodes a conserved major germline determinant[5,6]. A second mechanism takes place in the early embryo to complete mRNA localization, and involves mRNA decay and selective stabilization in the germ plasm[7]. The molecular basis underlying the link between mRNA decay and mRNA localization to the germ plasm has remained elusive.

Components of the Piwi-interacting RNA (piRNA) pathway, including Vasa and the PIWI protein Aubergine (Aub) are core components of germ granules[8], suggesting a potential link between the piRNA pathway and mRNA regulation in germ granules. piRNAs are a class of small 23–30 nucleotides (nt) RNAs bound to specific Argonaute proteins, the PIWI proteins. They are involved in the repression of transposable elements (TEs) in the germline[9,10]. piRNAs loaded into the cytoplasmic PIWI proteins Aub and Argonaute 3 (Ago3) target TE mRNA sequences through complementarity and guide their cleavage by the endonucleolytic activity of Aub and Ago3.

The more recent data have demonstrated the role of the piRNA pathway in post-transcriptional gene regulation[11–17]. In *Drosophila*, piRNAs produced in the female germline are provided maternally to the embryo, mostly loaded into Aub. Aub is present both in the somatic part of the embryo and at higher levels in the posterior germ plasm[15]. We previously showed that Aub targets maternal mRNAs through incomplete base-pairing with piRNAs and can induce their destabilization by either direct cleavage, or

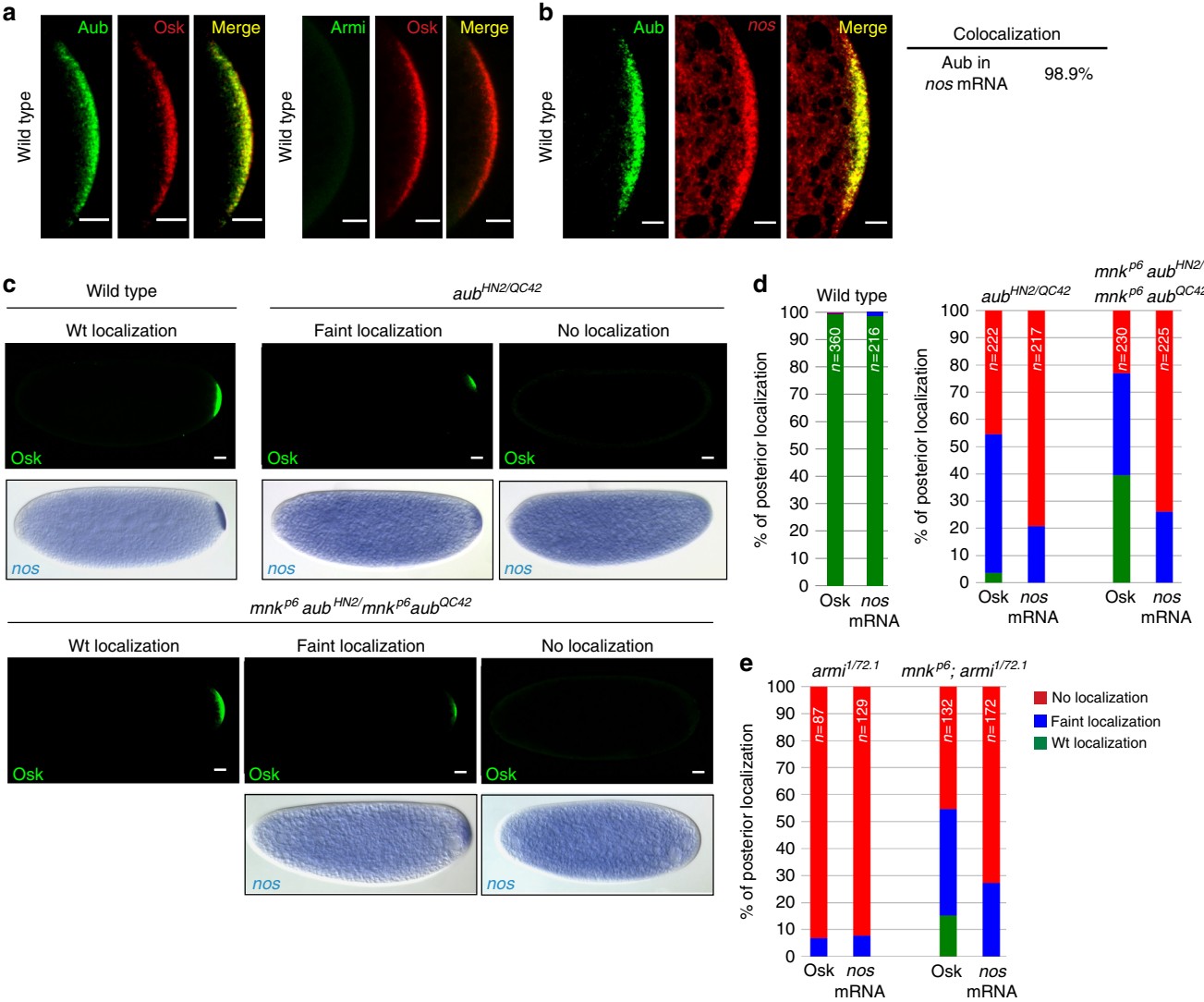

**Fig. 1** Aub and Armi have a direct role in *nos* mRNA posterior localization. **a** Immunostaining of wild-type 0–2 h-embryos with anti-Osk and either anti-Aub or anti-Armi. Posterior poles are shown. **b** Immuno-FISH of wild-type 0–2 h-embryos with anti-Aub and *nos* RNA probe. Quantification of colocalization using the Manders coefficient is indicated. Scale bars: 10 μm in **a**, **b**. **c** Immunostaining with anti-Osk and in situ hybridization with *nos* RNA probe of 0–2 h-wild-type, *aub* mutant or *mnk aub* double-mutant embryos. Scale bars: 30 μm. **d**, **e** Quantification of posterior localization of Osk and *nos* mRNA shown in **c** for *aub* and *mnk aub* mutant embryos, and in (Supplementary Fig. 1b) for *armi* and *mnk; armi* mutant embryos; the legend is in **e**. For all figures, the number of embryos (n) is indicated; the posterior pole is to the right

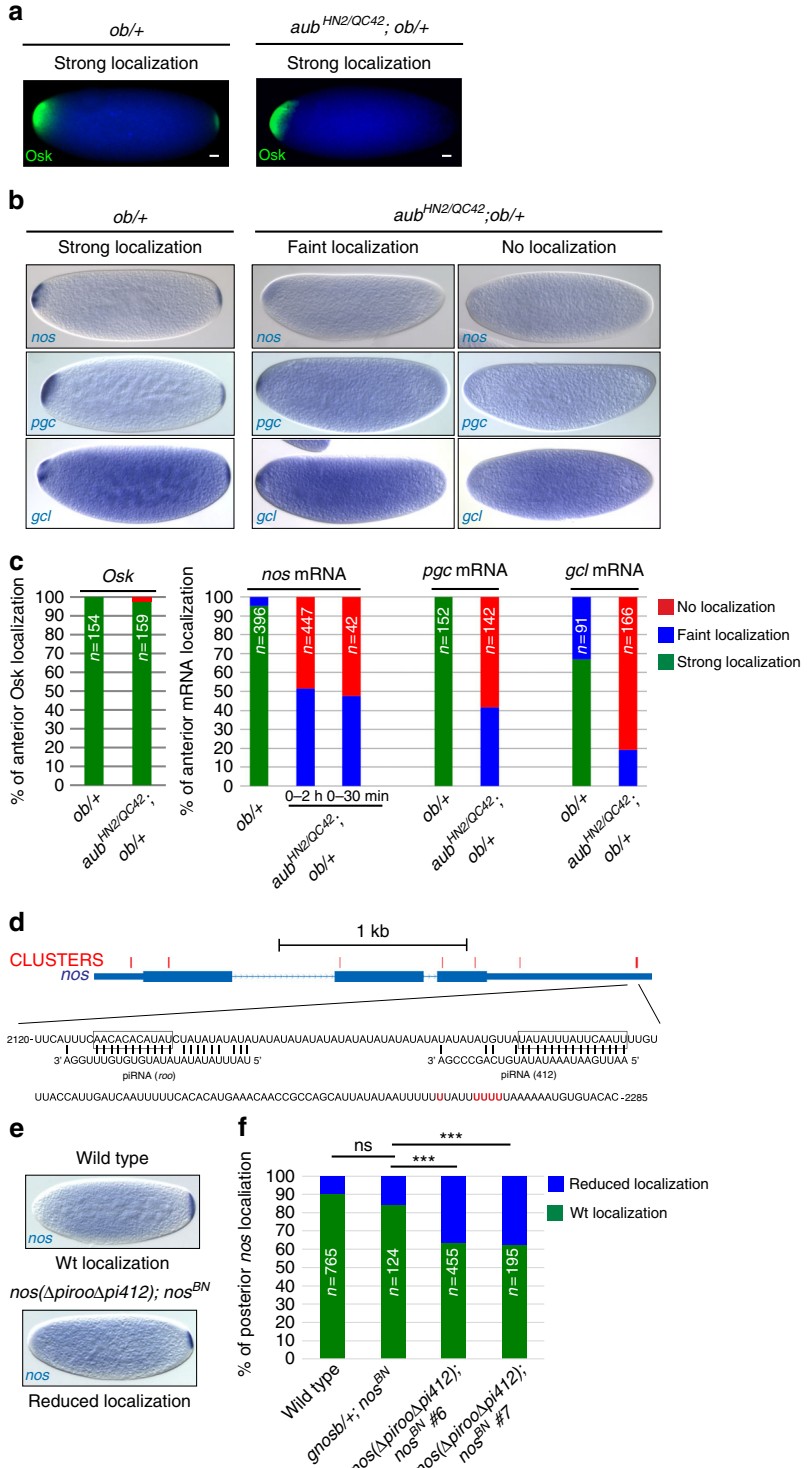

**Fig. 2** Role of Aub and piRNAs in germ cell mRNA localization to the germ plasm. **a** Immunostaining with anti-Osk of *osk-bcd3'UTR* (*ob*) embryos in wild-type or *aub* mutant backgrounds. The DAPI staining background (blue) shows the bulk of the embryo. Scale bars: 30 μm. **b** In situ hybridization of 0–2 h *ob/+* embryos either in wild-type or *aub* mutant backgrounds, with *nos*, *pgc* and *gcl* RNA probes. **c** Quantification of Osk and mRNA localization shown in **a**, **b**, respectively. 0–30min-embryos were also quantified for *nos* mRNA. **d** Schematic representation of *nos* mRNA and base-pairing with piRNAs. Thin boxes are 5'- and 3'-UTRs, lines are introns, and thick boxes are exons. Crosslink clusters from Aub-iCLIP are indicated in red. The sequence of the region with the strongest crosslink sites is shown. Base-pairing with representative piRNAs from *roo* and *412* TEs is shown; the deletions overlapping the piRNA target sites in the *nos(ΔpirooΔpi412)* transgene are boxed[15]. Aub-crosslinked nt are in red. **e** *nos* mRNA in situ hybridization of 0–2 h-embryos from wild-type and *nos(ΔpirooΔpi412)*; *nos^BN* females. The *nos^BN* mutant does not produce *nos* mRNA in the embryo. **f** Quantification of *nos* mRNA posterior localization as shown in **e**, for wild-type embryos, *nos^BN* embryos bearing the wild-type genomic *nos* (*gnosb*) transgene, and *nos^BN* embryos bearing the *nos*(*ΔpirooΔpi412*) transgene from two independent stocks. ns: non-significant, ***p < 0.001, using χ2 test

the recruitment of the CCR4-NOT deadenylation complex together with the RNA-binding protein Smaug (Smg)[11,15]. Strikingly, Aub-dependent unstable mRNAs that encode germ cell determinants, undergo selective stabilization: these mRNAs are degraded in the somatic part of the embryo, while they are stabilized and translated in the germ plasm to participate in germ cell development[11]. This raises the question of the role of piRNAs and Aub in the stabilization of these germ cell mRNAs in the germ plasm. Here we show, using *nos* mRNA as a paradigm, that piRNAs and Aub play an active role in the protection of germ cell mRNAs in the germ plasm.

Here we find that the germline-specific non-canonical poly(A) polymerase Wispy (Wisp) is a direct interactor of Aub, which colocalizes with Aub in the germ plasm. Furthermore using a method for poly(A) tail sequencing, we uncover the role of Aub in polyadenylation of a pool of germ cell mRNAs that have long poly(A) tails. We conclude that Aub acts by directly recruiting Wisp to germ cell mRNAs, leading to their polyadenylation and stabilization in the germ plasm. These results reveal a role for piRNAs in mRNA stabilization. They further identify a critical role of Aub and piRNAs in embryonic patterning through two opposite functions: somatic decay and germline stabilization of germ cell mRNAs, thus revealing the molecular link between these two processes.

## Results

### mRNA localization in germ plasm depends on Aub and piRNAs.

We have previously shown that Aub directly binds *nos* mRNA and is required for its deadenylation and decay in the somatic part of the early embryo[11,15]. However, *nos* mRNA also colocalizes with Aub protein in the germ plasm and primordial germ cells, where it is stabilized[18]. This suggests a different function of Aub in these regions. We used *aub* and *armitage* (*armi*) mutants to address a potential active role of Aub and piRNAs in *nos* mRNA localization in the germ plasm. Note that "mRNA localization" is used throughout, independently of the localization mechanism involved. Armi is another component of the piRNA pathway that has a prominent role in piRNA production[19]. We first validated the colocalization of Aub with *nos* mRNA in the germ plasm (Fig. 1a, b). Mutants of the piRNA pathway induce embryonic patterning defects through activation of the Chk2 DNA damage checkpoint. Checkpoint activation leads to defective localization of *osk* mRNA at the posterior pole of the oocyte and a lack of Osk protein synthesis[20,21]. This defect is partially rescued in double mutants for piRNA pathway components and the Chk2 kinase (*mnk* mutant)[20,21]. Accordingly, we found that most embryos produced by *aub^HN2*/*aub^QC42* females (referred to as *aub^HN2*/*aub^QC42* embryos) and all embryos produced by *armi^1*/*armi^72.1* females (*armi^1*/*armi^72.1* embryos) showed a lack or very weak localization of Osk protein at the posterior pole (Fig. 1c–e, Supplementary Fig. 1b). This defective germ plasm led to a very weak or an absence of *nos* mRNA localization at the posterior pole in these mutant embryos. Strikingly, a wild-type localization of Osk was rescued in 40 and 15% of *mnk aub*, and *mnk*; *armi* double-mutant embryos, respectively. However, wild-type *nos* mRNA posterior localization was not rescued (Fig. 1c–e, Supplementary Fig. 1a, b). This suggested a direct role of Aub in the localization of *nos* mRNA in the germ plasm. Armi did not localize to the posterior pole of the embryo (Fig. 1a), thus precluding a direct interaction between Armi and *nos* mRNA for *nos* localization at the posterior pole. Instead, strongly reduced piRNA levels in *armi* mutants[19] could underlie the lack of *nos* mRNA posterior localization in these mutants. Consistent with this, we previously showed that

unloaded Aub was unable to bind mRNAs and did not localize to the germ plasm[11].

To confirm a direct role of Aub in mRNA localization in the germ plasm, we used the *osk-bcd3'UTR* (*ob*) transgene containing the *osk* coding sequence followed by the *bicoid* (*bcd*) 3'UTR, which allows Osk localization and germ plasm formation at the anterior pole of the embryo[22]. Aub, as well as germ cell mRNAs were recruited and colocalized to this anterior germ plasm (Supplementary Fig. 2a)[11]. Although Osk localization to the anterior pole was not affected in *aub* mutant embryos bearing the *ob* transgene[23] (Fig. 2a, c), *nos* mRNA anterior localization was lost (48.5% of embryos) or very faint and diffuse (51.5% of embryos) (Fig. 2b, c). A similar result was obtained for two other germ plasm mRNAs that interact with Aub, *polar granule component* (*pgc*) and *germ cell-less* (*gcl*) (Fig. 2b, c), which is consistent with a direct role of Aub in the recruitment of germ cell mRNAs in the germ plasm. Interestingly, anterior localization of *nos* mRNA was not affected in *aub^-*; *ob/* + stage 10 oocytes, showing that Aub was not required at this stage (Supplementary Fig. 2b). The defects in *nos* mRNA localization were identical in 0–2 h- and 0–30 min-*aub* mutant embryos (Fig. 2c), indicating that Aub requirement for *nos* localization started between late oogenesis and early (0–30 min) embryogenesis.

Taken together, these results strongly suggest that Aub and piRNAs play a direct role in the localization of germ cell mRNAs in the germ plasm in late oocytes and/or early embryos.

### mRNA localization in germ plasm requires targeting by piRNAs.

Aub-iCLIP in embryos identified several reproducible sites of interaction between Aub and *nos* mRNA[11]. The most prominent crosslink sites were located in the distal region of *nos* 3'-UTR (Fig. 2d), and deletion of two putative piRNA target sites (from *roo* and *412* TEs) there led to defective *nos* mRNA deadenylation in the somatic part of the embryo[15]. We propose that the same piRNA-guided Aub interactions with *nos* mRNA result in deadenylation and decay in the soma, and localization in the germ plasm. We, thus, analyzed the localization of *nos* mRNA deleted for these two piRNA target sites. Posterior localization was significantly reduced in 36 and 37% of embryos in two independent *nos(ΔpirooΔpi412)*; *nos^BN* transgenic stocks, compared to localization of *nos* mRNA from a wild-type genomic transgene[24] (Fig. 2e, f). This localization defect was not due to reduced expression of the *nos(ΔpirooΔpi412)* transgene (Supplementary Fig. 2c). The fact that *nos* localization defect was weaker for *nos(ΔpirooΔpi412)* transgene than in *aub* mutant embryos, was expected since several Aub-binding sites remained unaffected in this transgene (Fig. 2d). Nonetheless, this reduced localization following the removal of only two piRNA target sites argues that Aub binding to at least a number of sites in mRNAs, depends on sequence-specific targeting by piRNAs. These results indicate that mRNA binding by Aub does not only rely on random piRNA targeting along the entire length of mRNAs through very low complementarity, as was proposed previously[25]. Consistent with this, using Aub-iCLIP data sets[11], we found that the proportion of mRNAs potentially targeted by piRNAs with high complementarity was significantly higher in Aub-crosslinked than in non-crosslinked mRNAs (Supplementary Fig. 2d). In addition, we found a strong overlap between Aub-crosslinked mRNAs and mRNAs producing piRNAs upon targeting by a highly complementary trigger piRNA[26] (Supplementary Fig. 2e).

These results are consistent with a role of piRNA sequence-specific targeting for Aub interaction with mRNAs and their localization to the posterior germ plasm.

**Aub methylation in both soma and germ plasm**. The dual role of Aub in mRNA decay in the soma and localization in the germ plasm indicates a switch in Aub function between these two regions of the embryo. Aub undergoes symmetric arginine dimethylation by the protein arginine methyltransferase 5, Capsuleen (Csul)[27]. Aub methylation is required for its interaction with Tudor (Tud) and its localization to the germ plasm[27,28]. We asked whether arginine dimethylation could participate in the switch in Aub function. In this hypothesis, only the pool of Aub localized in the germ plasm would be methylated, mRNA decay

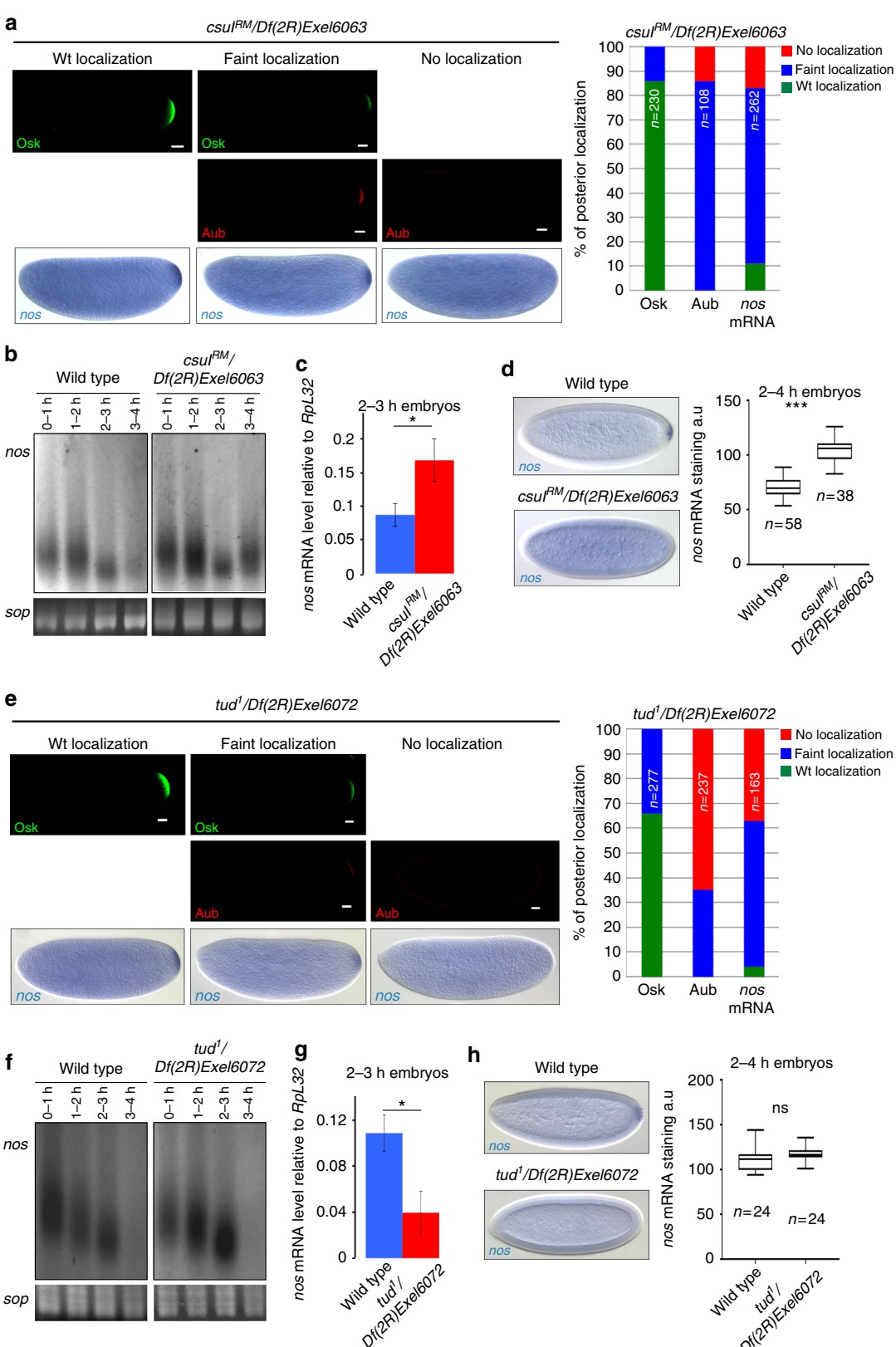

in the soma would involve unmethylated Aub. We analyzed both Aub functions when arginine dimethylation was defective in *csul* mutants. Posterior accumulation of both Aub and *nos* mRNA were strongly affected in *csul* mutant embryos (Fig. 3a, Supplementary Fig. 3a). We used poly(A) test (PAT) assays to record *nos* mRNA deadenylation in the bulk of *csul* mutant embryos during the first 4 h of embryogenesis. *nos* mRNA deadenylation profile was affected in mutant embryos, with long poly(A) tails still present at 3–4 h of development (Fig. 3b). This deadenylation defect correlated with stabilized *nos* mRNA in *csul* mutant embryos, both quantified by RT-qPCR and visualized in the soma by in situ hybridization (Fig. 3c, d). Consistent with impaired mRNA decay, a large proportion of *csul* mutant embryos did not hatch (Supplementary Fig. 3b).

Tud protein is restricted to the germ plasm in the embryo and is required for Aub posterior localization[29]. As expected, Aub and *nos* mRNA posterior localization was strongly affected in *tud* mutant embryos (Fig. 3e). In contrast, Aub-dependent somatic *nos* mRNA deadenylation and decay were not reduced in *tud* mutant embryos (Fig. 3f–h) and most of these embryos hatched (Supplementary Fig. 3b). Using embryo immunostaining with SYM11 antibody that specifically recognizes symmetric dimethylarginines, we confirmed the defect of arginine dimethylation in *csul* mutant embryos (Supplementary Fig. 3c). In *tud* mutant embryos, arginine dimethylation level was not reduced, but arginine dimethylated proteins did not accumulate in the germ plasm (Supplementary Fig. 3c).

These results indicate that both the somatic and germline pools of Aub are methylated. They further show that Aub arginine dimethylation is required for both Aub functions in mRNA decay and localization in the germ plasm. Therefore, this post-translational modification is not responsible for the switch in Aub function between soma and germ plasm.

**Aub recruits Wisp to stabilize mRNAs in the germ plasm**. Wisp is the germline-specific poly(A) polymerase involved in cytoplasmic polyadenylation of a large number of mRNAs during late oogenesis and early embryogenesis[30–34]. Since Wisp is required for stabilization and posterior localization of *osk* and *nos* mRNAs in embryos[30], we asked whether it could cooperate with Aub in mRNA stabilization and localization in the germ plasm. Wisp accumulated in the germ plasm in the oocyte where it strongly colocalized with Aub (Fig. 4a, b). Wisp was present in the whole embryo with higher accumulation in the germ plasm (Fig. 4b). It substantially colocalized with Aub in this region, as well as in primordial germ cells (Fig. 4b, Supplementary Fig. 4a). In addition, Wisp was recruited to the anterior germ plasm and colocalized with Aub in embryos expressing the *ob* transgene (Supplementary Fig. 4b). Analysis of Wisp-Aub colocalization in the germ plasm in *csul* and *tud* mutant oocytes showed that it was maintained in these mutant backgrounds, indicating that Wisp colocalization with Aub did not require Aub arginine dimethylation (Supplementary Fig. 4c). Strikingly, the levels of Wisp in the germ plasm correlated with the lower levels of posteriorly

localized Aub in these mutants, which is consitent with a role of Aub in the recruitment of Wisp to the germ plasm (Supplementary Fig. 4c). In contrast, the CCR4 deadenylase was depleted and Smg foci were smaller in the germ plasm where Aub accumulated (Fig. 4c).

We used co-immunoprecipitation in 0–2 h-embryos to show that Wisp coprecipitated with Aub, independently of the presence of RNA (Fig. 4d). The reverse experiment confirmed the coprecipitation of Aub with Wisp, in the absence of RNA, in 0–2 h-embryos (Fig. 4e). Moreover, this coprecipitation was maintained in 0–2 h-*csul* and -*tud* mutant embryos (Fig. 4e), revealing that the Aub/Wisp complex was independent of Aub arginine dimethylation, and could form in the somatic part of the embryo, since the levels of localized Aub were low in these mutant embryos.

Direct interactions between Aub and Wisp were analyzed using GST pull-down experiments. Aub contains three domains specific of Argonaute proteins (PAZ, MID and PIWI) (Fig. 4f). In vitro-translated HA-tagged Aub(1–482), which contained the PAZ domain, bound to recombinant GST-Wisp(1–713) and GST-Wisp(702–1373), but not to GST-Wisp(11–547) or GST alone (Fig. 4f). Wisp recombinant proteins that interacted with HA-Aub(1–482) overlapped over the central region of Wisp; we thus used GST-Wisp(636–746) to validate that the central region of Wisp interacted with HA-Aub(1–482) (Fig. 4f). In contrast, HA-Aub(476–866) that contained the MID and PIWI domains did not bind to any of the GST-Wisp proteins (Fig. 4f). These results reveal direct interactions between the central part of Wisp and the N-terminal half of Aub.

We used PAT assays and mPAT, a method for digital PAT assays multiplexed for high-throughput sequencing, to address a role of Aub in poly(A) tail elongation of *nos* mRNA localized in the germ plasm. Aub is required for deadenylation of the vast majority of *nos* mRNA (96%) present in the somatic region of the embryo[15]. Therefore, we used *ob*-expressing embryos to increase the pool of *nos* mRNA localized in the germ plasm. PAT assays from these embryos at 0–1 h and 1–2 h of development identified two pools of *nos* mRNA with poly(A) tails of ~12–60 nt, and 80 to 130 nt, respectively (Fig. 5a). In the wild-type background, poly(A) tail length of the pool with shorter poly(A) tails decreased in size with time by deadenylation, as previously reported[7]. In *aub* mutant embryos expressing *ob*, deadenylation of the pool with shorter poly(A) tails was reduced, and the pool with longer poly(A) tails completely disappeared (Fig. 5a). mPAT, which uses Illumina MiSeq based sequencing to identify and quantify PAT amplicons, confirmed these effects on both pools of *nos* mRNA in *aub* mutant embryos (Fig. 5b). In addition, similar effects were observed for *pgc* mRNA, but not for the control mRNA *tim10*, which was not bound by Aub (Supplementary Fig. 5a, b). Because *nos* mRNA deadenylation in the somatic region was not strongly affected in *tud* mutant embryos, we sequenced *nos* poly(A) tails in these embryos using mPAT, to address a role of posteriorly localized Aub in *nos* mRNA polyadenylation in the germ plasm. Strikingly, *nos* mRNA poy(A) tails longer than 80 nt were specifically shortened in *tud* mutant embryos at both time points (0–1 h and 1–2 h), consistent with the function of Aub in the

---

**Fig. 3** Role of Csul methyltransferase and Tud for Aub functions in mRNA decay and localization. **a**, **e** Immunostaining with anti-Osk and anti-Aub, and *nos* mRNA in situ hybridization of 0–2 h-*csul* (**a**) and -*tud* (**e**) mutant embryos. Quantifications of posterior localization in embryos shown in **a**, **e** are indicated (right panels). Scale bars: 30 μm. **b**, **f** PAT assays of *nos* mRNA in embryos at 1 h-intervals up to 4 h of development in wild type, *csul* (**b**) and *tud* (**f**) mutants. *sop* was used as a control mRNA. **c**, **g** *nos* mRNA quantification using RT-qPCR in 2–3 h wild-type, *csul* (**c**) and *tud* (**g**) mutant embryos. Normalization was with *RpL32* mRNA. For each genotype, mRNA levels at 2–3 h were normalized to the levels at 0–1 h. Means are from three biological replicates. The error bars represent SD. *$p < 0.05$ using the two-tailed Student's $t$ test. **d**, **h** *nos* mRNA in situ hybridzation of 2–4 h wild-type, *csul* (**d**) and *tud* (**h**) mutant embryos. Box plots showing the quantification of *nos* mRNA stabilization in the somatic part of the embryo. The central horizontal bar represents the median. ns: non-significant, ***$p < 0.001$, using the two-tailed Student's $t$ test

germ plasm for *nos* polyadenylation (Fig. 5c). Similar effects were recorded on *pgc* mRNA, but not the control *tim10* mRNA, sequenced using mPAT (Supplementary Fig. 5c).

Taken together, these results identify the role of Aub in the germ plasm for the recruitment of Wisp to germ cell mRNAs and their polyadenylation.

## Discussion

A number of recent studies have reported the role of piRNAs in cellular mRNA regulation, in addition to their role in repressing transposable elements. mRNA regulation by piRNAs is required for sex determination in *Bombyx mori* and embryonic patterning in *Drosophila*[11,14,15]. Furthermore, piRNAs are also involved in

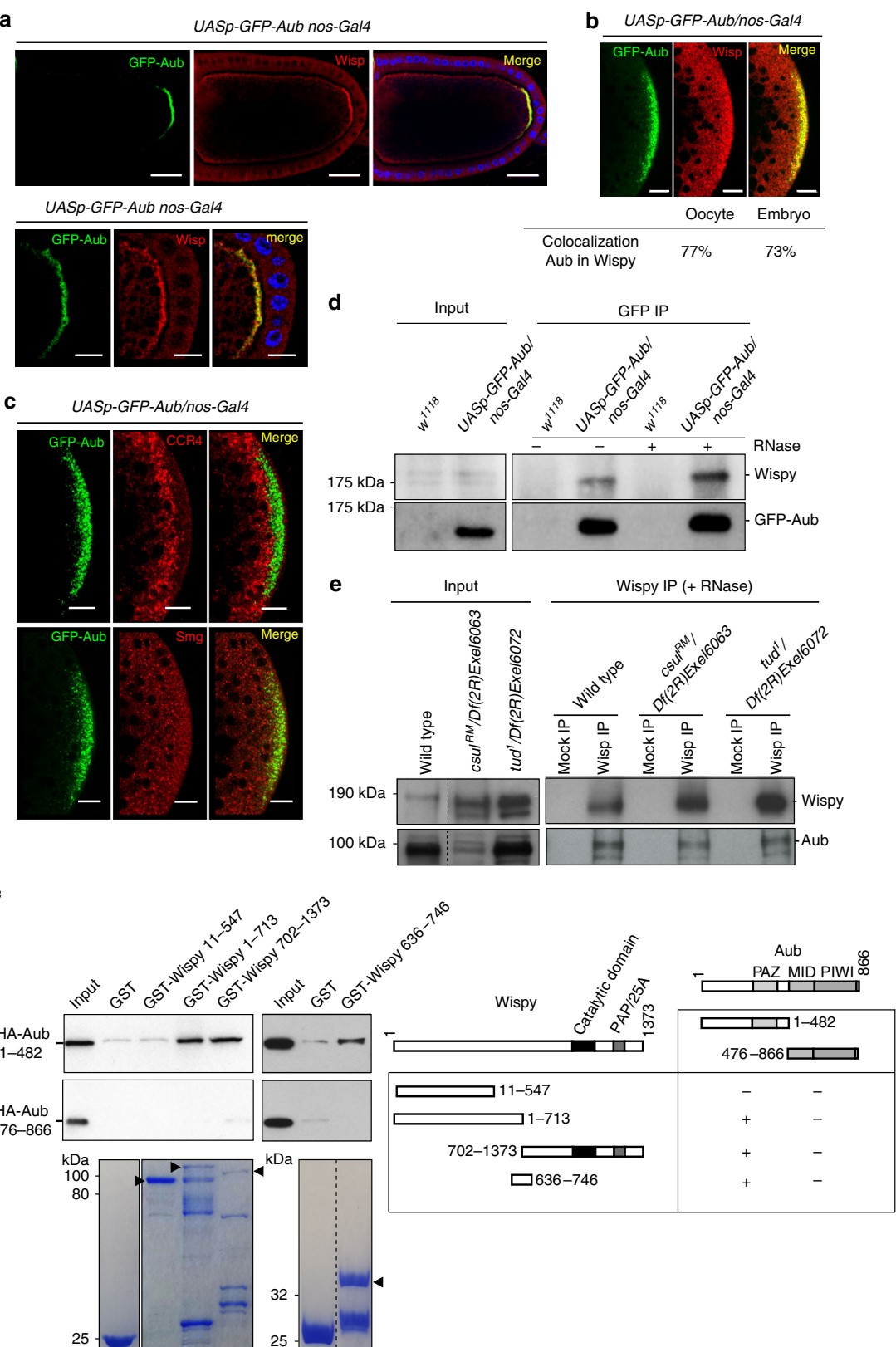

the decay of a large number of mRNAs and long non-coding RNAs during mouse sperm development[13,16]. mRNA base-pairing with piRNAs leads to their decay either by cleavage through the endonuclease activity of PIWI proteins, or by deadenylation[11–13,15–17]. Cleavage of cellular transcripts upon piRNA targeting produces sense piRNAs from these transcripts by ping-pong and phasing, thus unambiguously identifying mRNAs subject to this regulation[12,26].

Here we identify a mechanism of piRNA-dependent regulation that results in mRNA stabilization. Aub interacts with several hundred maternal mRNAs and induces their decay during the maternal-to-zygotic transition in the embryo[11]. Importantly, Aub-dependent destabilized mRNAs are locally stabilized in the germ plasm and encode germ cell determinants. A link has been described between maternal mRNA decay and posterior locali-zation, in the embryo[35]. We propose that Aub binding to these mRNAs and its dual role in somatic decay and stabilization in the germ plasm might be the basis for this link (Fig. 5d). A role of Aub in germ cell mRNA posterior localization has been hypo-thezised based on Aub interaction with these mRNAs[25], and Aub has been reported to be a component of the *nos* mRNA locali-zation complex[36]. Here we decipher the molecular mechanisms underlying this Aub function.

We identify a key component of the switch in Aub function between soma and germline. Aub directly interacts with the Wisp poly(A) polymerase, thus allowing poly(A) tail elongation and stabilization of germ cell mRNAs in the germ plasm. Osk is likely a major actor of mRNP-complex remodeling in the germ plasm, leading to Aub functional switch and Wisp activity (Fig. 5d). Osk colocalizes with Aub and Tud in germ granules[6]. Osk also directly interacts with Smg and prevents Smg interaction with *nos* mRNA, thus precluding its deadenylation and decay in the germ plasm[7,37]. Consistent with this, we show that in the germ plasm, Smg granules undergo remodeling and CCR4 deadenylase is depleted. The role of Aub and Wisp in polyadenylation points to their function in mRNA localization through selective stabiliza-tion in the germ plasm. However, a role of Aub and Wisp in the localization mechanism involving posterior anchoring during late oogenesis is also possible since Aub interacts with mRNAs in ovaries[25]. In addition, defects in mRNA anchoring occur in *wisp* mutants, suggesting a potential role of Wisp or mRNA poly(A) in anchoring[30].

The presence of Aub and Wisp in the same complex in *tud* mutant embryos, in which Aub localization in the germ plasm is very low, suggests the possible interaction of both proteins in the somatic region. Thus, Wisp might be present in a complex con-taining Aub, Smg and CCR4-NOT in the soma; however, its activity would be repressed by other components in this complex. This repression would be relieved in the germ plasm through the presence of Osk and the loss of Smg from the complex (Fig. 5d).

Like all members of the GLD-2 family of poly(A) polymerases, Wisp does not bind RNA but relies on RNA-binding proteins for its recruitment to mRNAs[38]. We identify Aub as an RNA-binding

protein involved in Wisp interaction with mRNAs in embryos. Intriguingly, Aub mode of mRNA binding that depends on diverse piRNAs is consistent with the lack of specific motifs in Wisp mRNA targets[32,34]. Poly(A) tail sequencing at the genomic scale has recently been used to identify mRNAs undergoing Wisp-dependent cytoplasmic polyadenylation in early embryos[33,34]. Comparison of these mRNAs to Aub-interacting mRNAs in embryos[11] showed that up to 25% of Wisp target mRNAs were also bound by Aub (Supplementary Fig. 5d), indicating a widespread role of Aub in Wisp-dependent cyto-plasmic polyadenylation.

Several studies addressing *nos* mRNA posterior localization reported the role of discrete, but partially redundant localization elements[4,36]. Our data are consistent with these findings: They propose the involvement of piRNA target sites highly com-plementary to piRNAs for mRNA localization. Deletion of these sites in *nos* mRNA induces only partial mislocalization, indicating redundancy with other localization elements.

These data reveal a role for piRNAs and PIWI proteins in mRNA stabilization, and uncover a major developmental func-tion of piRNAs in germ cell specification. They further highlight the central role of Aub in coupling piRNA inheritance and mRNA regulation for germ cell development and maintenance through generations.

## Methods

**Drosophila stocks and genetics**. The $w^{1118}$ stock was used as a control. Mutant stocks were $aub^{HN2}$ $cn^1$ $bw^1$/CyO, $aub^{QC42}$ $cn^1$ $bw^1$/CyO[39], $w^*$; $armi^{72.1}$/TM6C, $y^1$ $w^*$; $P\{lacW\}armi^1$/TM3[40], $mnk^{P6}$[41], $mnk^{P6}$ $aub^{HN2}$/CyO, $mnk^{P6}$ $aub^{QC42}$/CyO, $mnk^{P6}$; $armi^1$/SM6-TM6B, $mnk^{P6}$; $armi^{72.1}$/SM6-TM6B[20], $tud^1$ $bw^1$ $sp^1$/CyO[42], $w^{1118}$; Df(2 R)Exel6072/CyO that overlaps $tud$[43], $csul^{RM}$/CyO[44], $w^{1118}$; Df(2 R) Exel6063/CyO that overlaps $csul$[43], $nos^{BN}$/TM3[45], $y^1$ $wisp^{KG05287}$/FM7c[30] and Df(1) RA47/FM7c that overlaps *wisp*. Transgenic stocks were $osk$-$bcd3'UTR$ ($ob21$ and $ob42$ on second chromosome and $ob31$ on third chromosome)[22], $nos$-Gal4:VP16[46], UASp-GFP-Aub[47], nos($\Delta piroo$-$\Delta pi412$)[15], and $gnosb$ (wild-type genomic $nos$ transgene)[24].

**Immunostaining and RNA in situ hybridization**. Embryos were dechorionated with bleach for 3 min and thoroughly rinsed with $H_2O$. They were fixed in 37% formaldehyde with heptane (1:1) for 7 min on a wheel; formaldehyde was replaced by methanol and embryos were vortexed for 1 min. Embryos that sank to the bottom of the tube were rinsed three times with methanol. Before immunostaining, embryos were gradually rehydrated with methanol-PBT (PBS supplemented with 0.1% Triton-X 100) and washed three times with PBT. Embryos were incubated on a wheel at room temperature twice for 30 min in PBT, once for 20 min in PBT 1% BSA, and at 4 °C overnight in PBT 1% BSA with primary antibodies. Embryos were rinsed three times, washed twice for 30 min in PBT, then incubated in PBT 1% BSA for 30 min, and in PBT 1% BSA with secondary antibodies for 2 h at room tem-perature. Embryos were rinsed three times and washed twice for 30 min in PBT. Ovaries were dissected at room temperature in PBS, fixed with 4% paraf-ormaldehyde, rinsed and blocked with PBT containing 1% BSA for 1 h, and incubated in PBT 1% BSA with primary antibodies overnight at 4 °C. Ovaries were washed three times in PBT 1% BSA for 10 min at room temperature. They were incubated in PBT 0.1% BSA with secondary antibodies for 2 h at room tempera-ture, then washed three times in PBT for 10 min. DNA staining was performed using DAPI at 0.5 μg mL$^{-1}$. Primary antibody dilutions for immunostaining were mouse anti-GFP (Roche IgG1κ clones 7.1 and 13.1) 1:200; rabbit SYM11 antibody (EMD Millipore, 07-413) 1:200; rabbit anti-Osk (a gift from P. Lasko) 1:1000;

**Fig. 4** Wisp colocalizes and interacts with Aub. **a**, **b** Immunostaining of *UASp-GFP-Aub nos-Gal4* stage 10 oocytes, also stained with DAPI (*blue*) (**a**) and 0–2 h-embryos (**b**), with anti-GFP and anti-Wisp. Posterior poles are shown in the bottom panels in **a**, and in **b**. Quantification of colocalization in the germ plasm using the Manders coefficient is shown in **b**. Scale bars: 30 μm in **a** top panels, and 10 μm in **a** bottom panels and in **b**. **c** Immunostaining of *UASp-GFP-Aub/nos-Gal4* 0–2 h-embryos with anti-GFP and either anti-CCR4 (top panels), or anti-Smg (bottom panels). Scale bars: 10 μm. **d** Co-immunoprecipitation of Wisp with GFP-Aub in *UASp-GFP-Aub/nos-Gal4* 0–2 h-embryos. $w^{1118}$ 0–2 h-embryos were used as negative control (Mock). Immunoprecipitation was with anti-GFP (GFP IP) either in the presence (+) or the absence (−) of RNase A. **e** Co-immunoprecipitation of Aub with Wisp in wild-type, *csul* and *tud* mutant 0–2 h-embryos. Immunoprecipitation was with anti-Wisp in the presence of RNase A. Bound proteins were detected using western blots with anti-Wisp and anti-Aub; inputs correspond to protein extracts before IP in **d**, **e**. **f** GST pull-down assays between GST-Wisp and HA-Aub. Constructs and interactions are shown in the table. HA-tagged Aub fragments were revealed using western blot with anti-HA. Inputs correspond to 1:10 of in vitro-synthetized HA-Aub fragments before pull-down. GST alone was used as a negative control. GST and GST-recombinant proteins used in each pull-down are shown (bottom panels). Arrowheads indicate full-length recombinant proteins

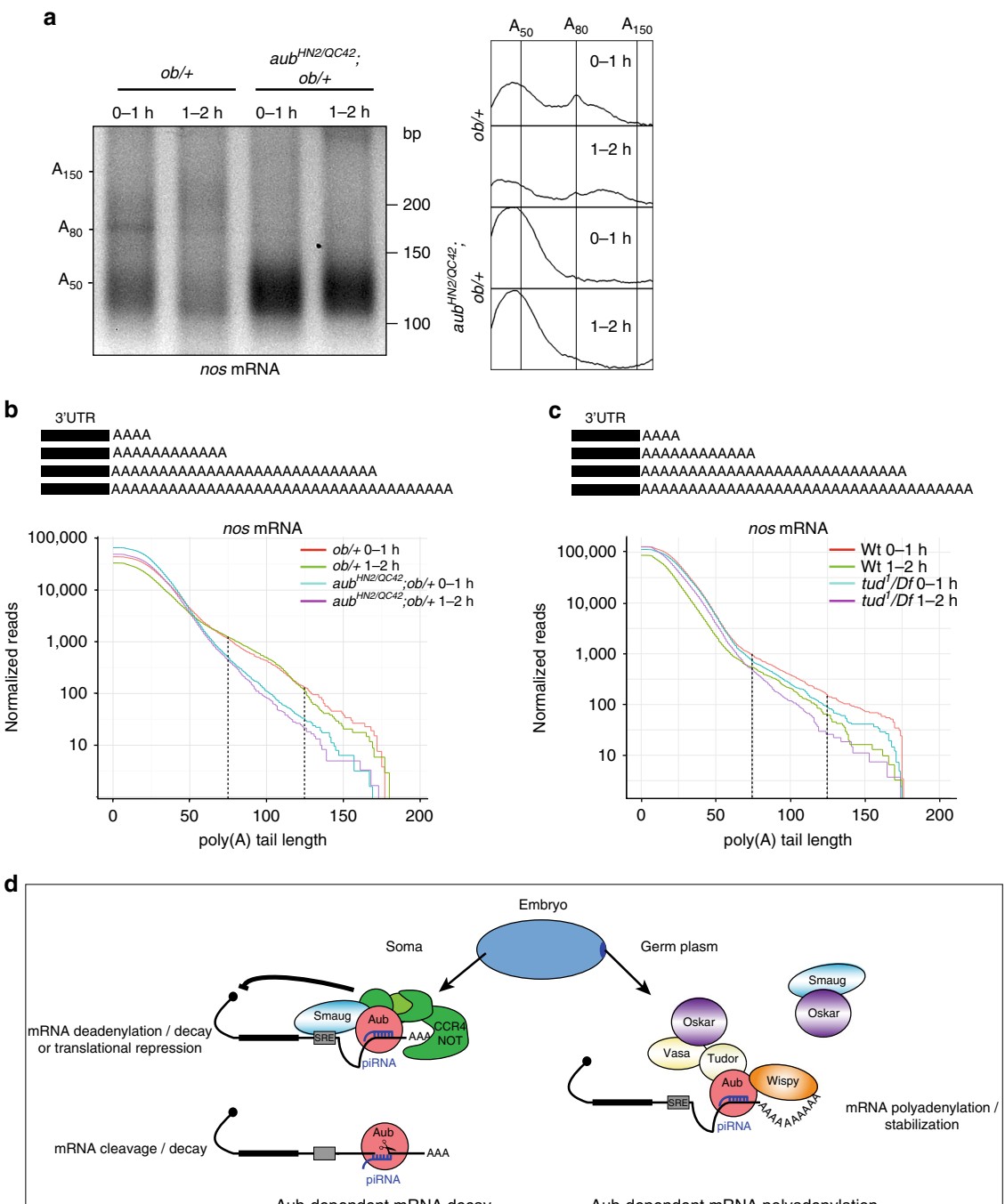

**Fig. 5** Aub recruits Wisp to stabilize germ cell mRNAs in the germ plasm. **a** ePAT assays of *nos* mRNA in 0–1 h- and 1–2 h-*ob/+* embryos, in wild-type and *aub* mutant backgrounds. ePAT assay profiles using ImageJ are shown on the right. **b**, **c** Distribution of sequenced *nos* mRNA poly(A) tails in *ob/+* and *aub⁻*; *ob/+* mutant embryos (**b**), and in wild-type and *tud* mutant embryos (**c**). Each curve represents the mean of two biological replicates normalized to reads per million. The mPAT reads are displayed as cumulative plots: all reads having at least a specific number of non-templated A-bases are pooled, as shown in the scheme (top panel). The *x*-axis represents the number of non-templated A-bases sequenced at the end of each read and the *y*-axis represents the normalized number of reads on a log scale. The portion of the graph corresponding to the pool of *nos* mRNA with long poly(A) tails is indicated with broken lines. In **b**, the proportion of reads having 100 A-bases or more (% of total) is significantly reduced in the *aub* mutant ($p = 0.0025$) by two-way ANOVA, which takes into consideration both time and genotype variables. **d** Model of the dual function of piRNA-loaded Aub in maternal mRNA somatic decay and stabilization in the germ plasm. See text for details

mouse anti-Aub (4D10)[48] 1:800; rabbit anti-Aub (Abcam, ab17724) 1:100; mouse anti-Armi[49] 1:500; mouse anti-Wispy (7B3, a gift from N. Kim) 1:100; guinea pig anti-Smg[50] 1:1000; and rabbit anti-CCR4[51] 1:200. Secondary antibodies (anti-rabbit Alexa 488-conjugated (Invitrogen, A-11034); anti-mouse Alexa 488-conjugated (Invitrogen, A-11029); anti-rabbit Cyanine 3-conjugated (Jackson Immunoresearch, 711-165-148); anti-mouse Cyanine 3-conjugated (Jackson

Immunoresearch, 115-166-006); anti-mouse Alexa 647-conjugated (Invitrogen, A-21236); anti-guinea pig Cyanine 3-conjugated (Jackson Immunoresearch, 706-165-148)) were used at dilution 1:500. For whole-mount in situ hybridization experiments, fixed embryos were rehydrated gradually with methanol-PBT-Tween (PBS supplemented with 0.1% Tween-20) and washed three times in PBT-Tween. Embryos were prehybridized in pre-warm HB buffer (50% formamide, 5× SSC, 50

µg mL$^{-1}$ heparin, 0.1% Tween-20, 5 µg mL$^{-1}$ torula yeast RNA (Sigma)) for 1 h at 65 °C and hybridized with 1:50 to 1:100 of probe in HB buffer overnight at 65 °C. Embryos were washed for 15 min at 65 °C twice with 2X SSC, 0.1% Tween-20, twice with 0.2× SSC, 0.1% Tween-20, and at room temperature three times with PBT-Tween. Embryos were incubated with preabsorbed anti-Digoxigenin-alcaline phosphatase antibody (Roche, 11093274910) at 1:2000 for 1.5 h at room temperature on a wheel. They were washed three times for 20 min in PBT-Tween and once for 20 min in AP buffer (100 mM Tris 2 M pH 9.5, 50 mM MgCl$_2$, 100 mM NaCl, 0.1% Tween-20). Revelation was in 1 mL AP buffer with 7.5 µL NBT (nitro blue tetrazolium) and 5.6 µL BCIP (5-bromo-4-chloro-3-indolyl-phosphate) for 5–15 min, and was stopped by three washes with PBT-Tween. Embryos were dehydrated gradually with ethanol-PBT-Tween and washed twice in ethanol 100%. Mounting was in Canada Balsam supplemented with methyl salicylate (4:1). Probes for in situ hybridization experiments were Digoxigenin-labeled antisense RNA in vitro transcribed from coding regions of the corresponding genes cloned into the Topo TA pCRTM II vector (Invitrogen)[11] or from the pN5 *nos* cDNA clone. For fluorescent in situ hybridization coupled with immunostaining (immuno-FISH), after hybridization and washing of the probe, embryos or ovaries were blocked 1 h in PBTHBR (1× PBS, 0.1% Triton-X100, 0.04% horse serum, 0.001% BSA, 40 U µL$^{-1}$ RNase Inhibitor (Promega)) and incubated overnight at 4 °C in PBTHBR with anti-Digoxigenin-POD antibody (peroxidase-conjugated, Jackson Immunoresearch, 200-032-156) at 1:200 and other antibodies. Embryos or ovaries were washed three times for 20 min in PBT-Tween. Tyramide Signal Amplification was performed at room temperature on a wheel for 10 min with 1:25 TSA Cyanine 3 in amplification diluent provided by the manufacturer (TSA® Cyanine 3, PerkinElmer). The reaction was stopped with two quick washes in PBT-Tween, followed by three 10 min-washes in PBT-Tween. Secondary antibodies were incubated at room temperature for 2 h and washed three times for 10 min in PBT-Tween. Mounting was in Vectashield.

**Microscopy and image processing.** Fluorescent images were acquired using a Zeiss LSM 780 laser scanning confocal microscope equipped with a Zeiss 40× PLAN-APO 1.3 oil-immersion DIC (UV) VIS-IR and a Zeiss 20× PLAN-APO 0.8 objective lens. The acquisition software was Zen. Contrast and relative intensities were processed and quantified with ImageJ software. Light microscope images were acquired using Leica Leitz DMRB Fluorescence-Phase Contrast Microscope with Nomarsky lens.

**RT-qPCR and PAT assays.** Total RNA was prepared from 30 embryos using Trizol (Invitrogen). RNA concentration was determined with nanodrop ND-1000 spectrophotometer. For RT-qPCR, 0.5–1 µg of total RNA was reverse transcribed with SuperScript III (Invitrogen) and random hexamers (Invitrogen). RNA levels were calculated using the LightCycler® 480 SYBR Green I Master (Roche) on the LightCycler® 480 Instrument (Roche) and the primers 5′-CGGAGCTTCCAA TTCCAGTAAC-3′ and 5′-AGTTATCTCGCACTGAGTGGCT-3′ for *nos*, and 5′-CTGTGAGAGTTCGCCAAATG-3′ and 5′-CATTGAGTTTCCGGTGTGTC-3′ for *RpL32*. Poly(A) test (PAT) assays were performed with 1 µg of total RNA using either regular PAT (Fig. 3), or ePAT (Fig. 5) methods[52]. For the PAT reaction, mRNA poly(A) tails were coated with 0.1 µg oligo-d(T)$_{12-18}$ primers which were then ligated with 40 U of T4 DNA Ligase; this reaction was followed by annealing of the d(T)-anchor primer to the overhanging remaining As at 12 °C and its subsequent ligation, then by reverse transcription with SuperScript III (Invitrogen) from this ligated primer, and PCR using the primers 5′-GCGAGCTCCGCGGCCG CGTTTTTTTTTTTT-3′ (d(T)-anchor) and 5′-TTTTGTTTACCATTGATCAATT TTTC-3′ for *nos* or 5′-GGATTGCTACACCTCGGCCCGT-3′ for *sop*. For ePAT, mRNA poly(A) tails was annealed with the d(T)-anchor primer by mixing 1 µg of total RNA with 2 µL of 50 µM d(T)-anchor primer and used as template for mRNA extension with 5 U of DNA Polymerase I, Large (Klenow) Fragment (New England Biology) at 37 °C; this reaction was then switched to 55 °C to dissociate annealings that had not been extended by DNA Polymerase I, and followed by reverse transcription with SuperScript III (Invitrogen) and PCR using d(T)-anchor and the specific primer 5′-GAAAAATTCAATGGCTCGAGTGCC-3′ for *nos*. PCR fragments were visualized on 2% agarose gel.

**mPAT.** To improve the resolution and sensitivity of gel-based PAT assays, we adapted the ePAT approach[53] to multiplexing on the Illumina MiSeq instrument. Sequencing ensures that any amplicon detected is specific to the gene of interest, and enables a digital read-out of the amplicon amount, and visualization of the distribution of sequenced poly(A) lengths. We refer to this assay as mPAT for multiplexed Poly(A) Test. A nested-PCR approach was used to sequentially incorporate the P5 and P7 elements necessary for bridge-amplification and sequencing on the Illumina flow-cell. First, ePAT cDNA was generated using the mPAT Reverse Primer 5′-CAGACGTGTGCTCTTCCGATCTTTTTTTTTTTTTTT-3′ using 500 ng total RNA from the indicated genotypes as input. In a first round of PCR amplification the sequence 5′-CCTACACGACGCTCTTCCGATCT-3′ was appended upsteam of traditional PAT primers designed ~100 nt from the polyadenylation site: *nos* mPAT 5′-CCTACACGACGCTCTTCCGATCTCACACATG AAACAAACCGCCA-3′; *pgc* mPAT 5′-CCTACACGACGCTCTTCGATCTCAAG AACAAGGAGGGAAGCTCG-3′; *tim10* mPAT 5′-CCTACACGACGCTCTTCCG

ATCTGCGCTACGATTGTTAGAGGTAC-3′. A pool of such gene-specific primers were used in five cycles of first-round amplification with the mPAT reverse primer. Unincorporated primers from this first round amplification were removed using NucleoSpin columns (Macherey-Nagel). Eluted amplicons were entered into a second round PCR using the universal Illumina Rd1 sequencing Primer 5′-AAT GATACGGCGACCACCGAGATCTACACTCTTTCCCTACACGACGCTCTTCC G-3′ and TruSeq indexed reverse primers from Illumina, with ten cycles of amplification. Note that each experimental condition was amplified separately in the first round with identical pooled primers. In the second round, each experimental condition received a different indexing primer. These second-round PCR reactions were pooled, cleared of excess primers using AMPure XP beads (Beckman Coulter) and sequenced using the MiSeq Reagent Kit v2 with 300 cycles (i.e., 300 bases of sequencing) according to the manufacturer's specifications. The data were analyzed using established bioinformatics pipelines[54] and figures were generated using the R framework.

**Immunoprecipitations.** For immunoprecipitations, 0–2 h-embryos (≈100 µL embryos per IP) were homogenized in 500 µL DXB-150 (25 mM Hepes-KOH pH 6.8, 250 mM sucrose, 1 mM MgCl$_2$, 1 mM DTT, 150 mM NaCl, 0.1% Triton X-100) containing cOmplete™ EDTA-free Protease Inhibitor Cocktail (Roche) and either RNase Inhibitor (0.25 U µL$^{-1}$, Promega) or RNase A (2 µg mL$^{-1}$, Sigma). A total of 50 µL Dynabeads protein A (Invitrogen) were incubated with either 10 µL mouse anti-GFP (monoclonal antibody 3E6, Invitrogen, A-11120), 5 µL mouse anti-Wisp (14D1, a gift from N. Kim), or 5 µL purified mouse IgG (Invitrogen, 02-6502) (mock IP) for 1 h on a wheel at room temperature. Protein extracts were cleared on 30 µL Dynabeads protein A previously equilibrated with DXB-150 for 30 min at 4 °C. The pre-cleared protein extracts were incubated with Dynabeads protein A bound to antibodies for 3 h at 4 °C. The beads were then washed 7 times with DXB-150 for 10 min at room temperature. Proteins were eluted in 1X NUPAGE buffer supplemented with 100 mM DTT at 70 °C and analyzed using western blots with antibodies at the following dilutions: mouse anti-Aub (4D10)[48] 1:2500; rabbit anti-Aub (Abcam, ab17724) 1:2000; guinea pig anti-Wispy[30] 1:3000; rabbit anti-Wispy[31] 1:2500. Complete blots are shown in Supplementary Fig. 6.

**GST pull-down assays.** The constructs for production of GST-Wisp(11-547) and GST-Wisp(702-1373) were previously generated[30]. The N-terminal half of Wisp (amino acids 1–713) was cloned into the pGEX-5X-2 vector using *Not*I and *Xho*I. The central region of Wisp (amino acids 636–746) was amplified by PCR and cloned into the pGEX-4T-1 vector, digested with *Eco*RI and *Xho*I. HA-Aub constructs were obtained by cloning PCR-amplified Aub fragments (amino acids 1–482 and 476–866) into the *Eco*RI and *Xho*I sites of the pCSH2 vector (pCS2 + backbone vector with two HA tags). GST-fused proteins were expressed in *E. coli* BL21 and affinity-purified on glutathione-Sepharose 4B beads (GE Healthcare); the beads were incubated overnight at 4 °C in PBT, cOmplete™ EDTA-free Protease Inhibitor Cocktail (Roche) and 5% BSA. HA-tagged proteins were synthesized in vitro using the TnT Coupled reticulocyte lysate system (Promega). HA-tagged proteins were incubated with immobilized GST fusion proteins for 1.5 h (45 min at room temperature followed by 45 min at 4 °C) in 400 µl binding buffer (50 mM Hepes pH 7.5, 600 mM NaCl, 0.2 mM EDTA, 1 mM DTT, 0.5% Nonidet P-40, cOmplete™ EDTA-free Protease Inhibitor Cocktail (Roche)) containing 0.2 µg µL$^{-1}$ RNase A. Beads were washed four times with binding buffer. Recombinant proteins were dissociated from the beads by boiling 5 min in Laemmli buffer and separated on a SDS-PAGE gel. Western blots were revealed with mouse anti-HA antibody (Covance, MMS-101R) at dilution 1:1000.

**Statistical and bioinformatic analyses.** Colocalization was analyzed using the ImageJ tool Coloc2 with 4–5 embryos or oocytes and calculated using the Manders's overlap coefficient[55]. Prediction of piRNA target sites on cellular mRNAs was performed as follows. We used a pool of piRNAs from 0–2 h-embryos sequenced in previously published libraries (GSM327625, GSM327626, GSM327627, GSM327628, GSM327629, GSM1818089, GSM1818091). This led to a total of 3,305,903 non-redundant piRNA sequences. Bowtie was used with different complementarities to identify piRNA target sites on transcripts with reproduced cross-links. Bowtie with option '-v 0', '-v1', '-v 2' or '-v3' was used to identify piRNAs that potentially target mRNAs with up to 0, 1, 2 or 3 mismatch (es), respectively. For complementarities with a seed, we did not use quality values, therefore the sum of the quality values at all mismatched read positions (-e/-maqerr) was set to an arbitrary value of 2000, which disabled the quality values. Furthermore, -l (length of the seed) and -n (number of mismatches within the seed) were set to different values. The option '–nofw' was used to search only for reverse-complementarity between piRNAs and mRNAs.

**Data availability.** Accession numbers of previously published datasets are GSM327625, GSM327626, GSM327627, GSM327628, GSM327629, GSM1818089, GSM1818091. mPAT sequences generated in this study have been deposited to figshare, under the link: https://doi.org/10.4225/03/59b074beaefcc

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

## Acknowledgements

This work is dedicated to the memory of our friend and colleague Cathy Papin. We thank P. Lasko, N. Kim, M. Siomi and M. Wolfner for their gifts of antibodies, and A. Ephrussi, E. Gavis, P. MacDonald and W. Theurkauf for their gifts of fly stocks. We thank M. Curtis, A. Barugahare, Micromon and the Monash Bioinformatic Platform for technical support. This work was supported by UMR9002 CNRS-University of Montpellier, ANR (ANR-2010-BLAN-1201 01 and ANR-15-CE12-0019-01), FRM ("Equipe FRM 2013 DEQ20130326534" and "Projets Innovants ING20101221078") and Fondation ARC (ARC Libre 2009, N°3192). J.D. was supported by Fondation ARC and ANR, G.B. by Fondation ARC and FRM and C.J. by the Labex EpiGenMed (ANR-10-LABX-12-01). S.P. held a salary from FRM "Projets Innovants" and "Equipe FRM 2013". T.H.B. was supported by a Biodiscovery Fellowship from Monash University.

## Author contributions

M.S. and C.P.: Conceived the study; J.D., G.B., A.C., C.J., A.-C.M., C.P., M.S.: Performed experiments and analyzed the data. S.P.: Performed bioinformatic analyses. T.H.B. and P.F.H.: Performed mPAT and mPAT bioinformatic treatment, respectively. M.S.: wrote the manuscript with assistance from J.D. All authors discussed the manuscript.

## Additional information

**Competing interests:** The authors declare no competing financial interests.

