## [Peer Review File · Nature Communications]

Reviewers' comments:

Reviewer #1 (Remarks to the Author):

The article by Dufort et al addresses the question of RNA localization in the germ plasm of *Drosophila*. The authors confirm many previously reported results, such as the role of Aub in RNA localization, the effect of Oskar and Tudor on Aub localization. This study follows a previously reported analysis by the same lab suggesting that piRNAs, bound by Aub, contribute to the degradation of somatic RNAs. In the present study the authors argue that Aub at the posterior pole has a second function to stabilize localized RNAs. This is apparently achieved by an interaction between Aub and the RNA polymerase *Wispy* and thereby contributes to the localization pattern.

Overall the study is well performed, however the amount of new information is limited. It also remains unclear whether the effects observed by the authors are limited to RNA stability or as mentioned in the discussion also to RNA localization. Both would contribute to the enrichment of RNAs in the germ plasm.

There are a number of conclusions drawn made in the text that seem to be not strongly supported by the data presented.

1) The authors need to differentiate between "localization" and "stabilization" and specify exactly, what they are measuring. Through the first half of the text, the authors indicate that the various mutations and deletions are affecting mRNA localization when in fact they could be affecting mRNA stability. To prevent confusion, the authors should state mRNA "enrichment" as this is what they are in fact measuring. The enrichment of mRNAs in the germ plasm is affected by two opposing forces; mRNA localization, which increases mRNA levels in the germ plasm, and prevention of mRNA degradation, which protects mRNAs in the germ plasm. The authors use localization and stability interchangeably, which is incorrect, especially since they are measuring enrichment and not either of the two processes directly.

2) A major finding made by the authors is that Aubergine does not localize mRNAs in the germ plasm. This finding is contrary to a recent finding by Mourelatos and colleagues in *Nature*, where they concluded that Aubergine alone is responsible for mRNA localization in the germ plasm (<http://www.nature.com/nature/journal/v531/n7594/abs/nature17150.html>). Their *mnk*, *aub* double mutant as well as their *aub* mutant clearly still binds *nanos*, *pgc* and *gcl*, albeit at lower levels. Clearly, *aub* is not necessary for mRNA localization of these mRNAs. The lower levels, as the authors in the end suggest, could be due to destabilization of these mRNAs.

3) A very small % of RNA is not posteriorly enriched in *nosDelta piroodeltapi412*, when potential 'protective' pi RNA sequences are deleted. Deletion experiments of the 3'UTR of *nanos* by Gavis et al many years ago suggested that these sequences were not required for RNA localization. The present data may suggest that these sequences have a more specific role in RNA stability. However, as the effects of Aub mutants on *nos* RNA enrichment is so much stronger, it is not clear if Aub is only required for stability and may also contribute to localization.

4) Fig 2e *nosDelta piroodeltapi412* needs to have an appropriate control, which would be a wild-type transgene inserted at same genomic site and localization in *nos* mutant background needs to be assessed and compared for this control transgene.

5) Ping pong data are for entire embryo and are not conclusive for posterior localization as the germ plasm-localized *nos* RNA is only 4% of total. So it is unclear if this result is relevant to assess polyA tails in germ plasm, or more relevant to previous data by the authors where they described somatic degradation.

6) Polyadenylation should depend on the ability of Aub to interact with piRNA recognition within

nos RNA, so polyadenylation of nosDeltapiroodeltapi412 should be tested.

7) Csu1 mutants affect both RNA degradation and localization, so a specific role for Aub methylation and RNA localization cannot be established. The finding that in tud mutants RNA degradation is not affected could have allowed the authors to demonstrate specificity if changes in polyadenylation dependent on posterior localized Aub.

8) The authors show that upon Aubergine deletion the population of nanos mRNA (and general mRNAs) with a shorter polyA tail is markedly increased, indicating that Wispy is no longer able to adenylate the shortened nanos poly A tail. This result somehow contradicts the authors' previous work that showed that Aubergine causes deadenylation of nanos mRNA thereby stimulating its decay. In the absence of aub therefore, there should be no nanos in the embryo even at the posterior pole. How can the authors exclude Wisp:Aub interaction in the soma and stimulating the adenylation function of Aub on nanos mRNA in the soma?

9) Is the Aub-wispy interaction required for polyadenylation marks seen in wildtype embryos?

Minor points

1. While it is important and nice to see that data by others can be recapitulated, the amount of new results displayed in the major figures in this manuscript is less than 1/2 of total data in the manuscript. Many results shown in this manuscript are recapitulation of previous published data some even in the main figures. i.e.: Fig 1 and most of Fig 2 was already reported previously (Macdonald, Simonelig and others) The fact that csu1 mutants affect nos RNA localization is already known (Mechler). The effect of Tud on aub localization had already been demonstrated (Mourelatos, Lehmann, Xu).

2. Handler et al., 2011 (EMBO J) demonstrated and corrected previous results and showed that Tud is not part of the piRNA pathway as the authors state in their introduction.

Together, the reported findings and the conclusions seem somewhat preliminary. It would be great if the authors could provide more direct evidence to distinguish between a role of Aubergine in localization versus protection of stability or both.

The idea that germ plasm tethered polyadenylation and protection from degradation contributes to the enrichment of localized RNAs is attractive and adds to similar conclusions made by Lipshitz for the hsp83RNA. Lipshitz and colleagues showed that hsp83RNA is enriched in the germ plasm because of lack of degradation but independent of localization. Indeed, this RNA may be a good target to study the relative role of Aub enrichment due to lack of degradation separate from localization. Some additional data along these lines could substantially extend the conclusion drawn in the manuscript.

Reviewer #2 (Remarks to the Author):

The manuscript by Dufourt et al. reports a new mechanism of Aub/piRNA-dependent mRNA stabilization in the germ plasm of the Drosophila embryo. This mechanism is opposite to the known Aub-dependent destabilization of mRNAs in the somatic part of the embryo. The authors showed that Aub achieves the stabilization function in the germ plasm by directly interacting with the Wisp poly(A) polymerase that elongates poly(A) tail and stabilizes target mRNAs. The experiments are well designed and the results are overall convincing. The findings reported in this manuscript are exciting, with general significance to developmental biology and RNA biology. I support the publication of the manuscript in Nature Communication without major revision.

Before the manuscript is accepted, I suggest a minor revision to address the following points:

(1) P 4, ll 2-3: "Aub-dependent unstable mRNAs encode germ cell determinants that undergo selective stabilization:" This sentence should be revised because it is the mRNAs, but not germ cell determinants, undergo selective stabilization.

(2) Fig 1d: Show the percentage of Osk and nos mRNA in WT embryos with respect to normal, faint and no localization.

(3) P 6, para 2: "Taken together, these results show that Aub and piRNAs play a direct role in the localization of germ cell mRNAs in the germ plasm in late oocytes and/or early embryos." This is over-stated. Up to now, the authors have only shown the role of Aub, but not piRNA, in the germ granule localization.

(4) P 6, para 3, ll 9-11: "These results argue that Aub binding to mRNAs depends on sequence-specific targeting by piRNAs, rather than on random targeting along the entire length of mRNAs through very low complementarity, as was proposed recently". This is over-concluded as well, since the localization of nos mRNA was reduced in only 36%- 37% of embryos deleted for both piRNA target sites. Hence, it still remains possible that some other piRNA sites, possibly in a semi-random fashion, are also required for the localization. Relevantly, the Aub-iCLIP data in Suppl Fig 2C are not very robust.

(5) P 9, Discussion: The role of piRNAs in cellular mRNA and lncRNA regulation in mammalian systems (ref. 16) should be mentioned here alone with silkworm and Drosophila studies to provide a more intellectually complete picture.

Reviewer #3 (Remarks to the Author):

Although the conventional wisdom for the regulatory function of the Piwi pathway is transposon and gene silencing, a body of literature also implicates the Piwi pathway in an RNA localization role for germ plasm RNAs. Since some Argonaute proteins in nematodes have genetic roles in promoting gene expression, the notion that Piwi proteins could stabilize or promote gene expression is a very intriguing possibility that has not yet been fully described in other animals. The Simonelig lab has carried out previous studies of Aubergine-targeted germ plasm RNAs that are subjected to deadenylation during early embryogenesis, which follows the conventional gene silencing/destabilization role for Argonaute/Piwi proteins.

In this study, the authors now examine a potential RNA stabilization role for Aubergine in the germ plasm that is located at the posterior pole of the oocyte/embryo. Whereas Osk protein serves as an Aub-independent posterior germ plasm localization control, Nanos (nos) mRNA localization at the posterior pole is disrupted in Aub and Armi mutants. Interestingly, although mnk double mutants can partially rescue Osk posterior localization, no rescue is observed for nos mRNA.

Since Aub localization has been previously connected to arginine methylation by Csil and Tud, these mutations are examined for Aub protein and nos mRNA localization. The data in Fig 3 is very nice but restricted to just Csil, so I found it a bit odd that the Tud data is relegated to Sup Fig 3. A suggestion that seems logical to me is that the two mutants data could go together a single Figure 3, even though it would be a complicated figure. However, I raise one major issue here:

Issue#1: On Pg 7, the authors propose the hypothesis that localized Aub at germ plasm is methylated while unmethylated Aub is in soma. But the only experiment assessing Aub arginine

diemethylation is showing in the csul mutant decreased posterior accumulation of Aub and nos mRNA and decreased nos mRNA deadenylation. However, Aub is also destabilized in the csul mutant, according to Kirino et al, 2009. The analysis of Tudor mutants however do not directly address arginine demethylation state even if it does affect Aub and nos mRNA localization but not nos mRNA deadenylation. What about embryo staining and western blots with the SYM11 antibody from csul and tud mutants to show that arginine diemethylation is unaffected while Aub and nos mRNA posterior localization is greatly affected? This would be the key experiment to allow the authors to speak to the hypothesis they had proposed.

In the last set of Figures 4 and S4, the authors test the interaction of Wispy, an embryonic PolyA polymerase, and demonstrate compelling Aub interactions. Finally, the authors conduct an mPAT assay that uses deep-sequencing to measure PolyA tail length for specific transcripts, and this is conducted in Aub and wispy mutant embryos. Here, I found three significant issues to raise:

Issue#2: The data of Fig 4a-d, the CoIP and colocalization of wispy with Aub very intriguing, but the study is still missing a test for whether these stainings and coIPs are disrupted in the tud or csul mutant backgrounds.

Issue#3: The mPAT assay cumulative distribution profiles need to have highlighting of which regions are significantly different between the time points and mutants vs genetic controls. The authors need to discuss and reconcile the inconsistency in the mPAT profiles with the traditional gel-based PAT assay, comparing Fig 4f and 4g. In particular, there is a concerning saturation of reads in the mPAT assay shorter than 50 PolyA's when there is no such saturation in the traditional PAT assay, and there are incongruency in the profiles. Marking the meaningful changes in distribution plots in Fig S4 is also needed.

Issue #4: The authors should conduct and describe an analysis on how much overlap in Aub iCLIP targets and mRNA targets deregulated in PolyA tails in the Wispy mutant, from Eichorn et al 2016 study? This should provide better scope as to how many of the overall Wispy targets are also subjected to Aub regulation.

Overall, I enjoyed reading this paper and found most of the data and story very interesting. I recommend the authors address all the main issues I raised in order to satisfy my full support of this study.

The referees' comments have been very helpful in strengthening the manuscript. We have addressed all their concerns in this revision, and the manuscript is now expanded by additional experiments in six Figures.

In this new version, several points have been reinforced and clarified. In particular, we addressed the impact of Aub arginine dimethylation on Aub colocalization and interaction with Wip1 poly(A) polymerase. These new results show that Aub methylation is not required for its interaction with Wip1. We also strengthened the role of posteriorly localized Aub in polyadenylation of localized mRNAs by sequencing poly(A) tails in *tudor* mutant embryos.

The point-by-point Response to the Referees is as follows.

Reviewer #1

The article by Dufort et al addresses the question of RNA localization in the germ plasm of *Drosophila*. The authors confirm many previously reported results, such as the role of Aub in RNA localization, the effect of Oskar and Tudor on Aub localization. This study follows a previously reported analysis by the same lab suggesting that piRNAs, bound by Aub, contribute to the degradation of somatic RNAs. In the present study the authors argue that Aub at the posterior pole has a second function to stabilize localized RNAs. This is apparently achieved by an interaction between Aub and the RNA polymerase Wip1 and thereby contributes to the localization pattern.

Overall the study is well performed, however the amount of new information is limited. It also remains unclear whether the effects observed by the authors are limited to RNA stability or as mentioned in the discussion also to RNA localization. Both would contribute to the enrichment of RNAs in the germ plasm.

There are a number of conclusions drawn made in the text that seem to be not strongly supported by the data presented.

1) The authors need to differentiate between "localization" and "stabilization" and specify exactly, what they are measuring. Through the first half of the text, the authors indicate that the various mutations and deletions are affecting mRNA localization when in fact they could be affecting mRNA stability. To prevent confusion, the authors should state mRNA "enrichment" as this is what they are in fact measuring. The enrichment of mRNAs in the germ plasm is affected by two opposing forces; mRNA localization, which increases mRNA levels in the germ plasm, and prevention of mRNA degradation, which protects mRNAs in the germ plasm. The authors use localization and stability interchangeably, which is incorrect, especially since they are measuring enrichment and not either of the two processes directly.

Thank you for bringing this point, we agree that it required clarification. Indeed, mRNA localization involves two additive mechanisms: diffusion and anchoring to the germ plasm during late oogenesis and selective stabilization in the germ plasm. We have now clarified this point in the Introduction p. 3, and included as a reference (Reference 2), the Review by Martin and Ephrussi (2009 Cell, 136, 719) that details these mechanisms. We further clarify in the Results section, p. 4 that we use the term "mRNA localization" independently of the localization mechanism involved. In addition, we have added a paragraph in the Discussion p. 12 regarding the potential localization mechanisms that depend on Aub.

2) A major finding made by the authors is that Aubergine does not localize mRNAs in the germ plasm. This finding is contrary to a recent finding by Mourelatos and colleagues in Nature, where they concluded that Aubergine alone is responsible for mRNA localization in the germ plasm (<http://www.nature.com/nature/journal/v531/n7594/abs/nature17150.html>). Their *mnk*, *aub* double mutant as well as their *aub* mutant clearly still binds *nanos*, *pgc* and *gcl*, albeit at lower levels. Clearly, *aub* is not necessary for mRNA localization of these mRNAs. The lower levels, as the authors in the end suggest, could be due to destabilization of these mRNAs.

As explained in Point 1, we have now included a paragraph about the possible roles of Aub in the different localization mechanisms (Discussion p.12). The publication by the Mourelatos group in 2016 (Nature 531, 390) proposes that the role of Aub in mRNA posterior localization acts in addition to other mechanisms (such as other protein-RNA interactions) (Discussion p. 393). Our results identify a new mechanism, but are not contradictory to their results.

3) A very small % of RNA is not posteriorly enriched in *nosDelta piroodeltapi412*, when potential 'protective' piRNA sequences are deleted. Deletion experiments of the 3'UTR of *nanos* by Gavis et al many years ago suggested that these sequences were not required for RNA localization. The present data may suggest that these sequences have a more specific role in RNA stability. However, as the effects of Aub mutants on *nos* RNA enrichment is so much stronger, it is not clear if Aub is only required for stability and may also contribute to localization.

We believe that the difference between the data described in Gavis et al. (Dev. Biol. 1996, 176, 30) and ours, might be explained by the fact that we record weaker defects. In addition, the approaches are different as we used deletions within a *nos* transgene, whereas Gavis et al. used either *lacZ* constructs with various part of *nos* 3'UTR, or insertion of *nos* 3'UTR in addition to *tubulin* 3'UTR.

We now clarify that *nos* localization from the *nos(ΔpirooΔpi412)* transgene is expected to be less affected than in *aub* mutant embryos, since many Aub binding sites remain unaffected in this transgene. This is indicated in the Results section p. 7.

4) Fig 2e *nosDelta piroodeltapi412* needs to have an appropriate control, which would be a wild-type transgene inserted at same genomic site and localization in *nos* mutant background needs to be assessed and compared for this control transgene.

We included this control in Figure 2f and Supplementary Figure 2c. We recorded *nos* localization with a wild-type *nos* genomic transgene in the *nos^{BN}* mutant background and compared it to localization with the *nos(ΔpirooΔpi412)* transgene. Localization with the deleted transgene was significantly affected (Figure 2f). The insertion sites of these transgenes were not identical, therefore we quantified, using RT-qPCR, the levels of *nos* mRNA in embryos, for the deleted and wild-type transgenes to show that the localization defect was not due to low expression of the deleted transgene (Supplementary Figure 2c).

5) Ping pong data are for entire embryo and are not conclusive for posterior localization as the germ plasm-localized *nos* RNA is only 4% of total. So it is unclear if this result is relevant to assess polyA tails in germ plasm, or more relevant to previous data by the authors where they described somatic degradation.

We agree with this remark. However, as we propose in the Results section p.6, piRNA base-pairing to mRNAs would be the same for mRNA decay in the soma and localization in the germ plasm. The comparison in Supplementary Figure 2e is meant to show that a large proportion of Aub-interacting mRNAs are targeted by highly complementary piRNAs. Because this information is valid for both mRNA decay and localization, we conserved it.

6) Polyadenylation should depend on the ability of Aub to interact with piRNA recognition within nos RNA, so polyadenylation of nosDeltapiroodeltapi412 should be tested.

The localization defect with the *nos*(Δ *piroo* Δ *pi412*) transgene is modest (Result section p. 7): reduced localization in 37% or 36% of embryos. Therefore, we believed that recording a difference in poly(A) tail on this very low level of mRNA (4% of *nos* mRNA in 36% of embryos) was not technically possible. Nonetheless, we did sequence poly(A) tails in both *nos*(Δ *piroo* Δ *pi412*) transgenic stocks, at the two time points. As expected no difference with poly(A) tails of wild-type embryos could be recorded.

7) *Csul* mutants affect both RNA degradation and localization, so a specific role for Aub methylation and RNA localization cannot be established. The finding that in *tud* mutants RNA degradation is not affected could have allowed the authors to demonstrate specificity if changes in polyadenylation dependent on posterior localized Aub.

We performed the sequencing in *tud* mutant embryos for *nos*, *pgc* and the control *tim10* mRNAs. These experiments are presented in Figure 5c and Supplementary 5c. They show reduced levels of the pool of *nos* and *pgc* mRNAs with long poly(A) in *tud* mutants at both time points, and therefore confirm the role of localized Aub in polyadenylation.

8) The authors show that upon Aubergine deletion the population of nanos mRNA (and general mRNAs) with a shorter polyA tail is markedly increased, indicating that Wispy is no longer able to adenylate the shortened nanos poly A tail. This result somehow contradicts the authors' previous work that showed that Aubergine causes deadenylation of nanos mRNA thereby stimulating its decay. In the absence of aub therefore, there should be no nanos in the embryo even at the posterior pole. How can the authors exclude Wisp:Aub interaction in the soma and stimulating the adenylation function of Aub on nanos mRNA in the soma?

The data in Figure 5a and b are consistent with our previous data on the role of *aub* on deadenylation of the large pool of *nos* mRNA present in the soma. This pool has short poly(A) tails (up to 65A) and is deadenylated with time. Aub is required for this deadenylation. We see both on the gel in Figure 5a, and with poly(A) tail sequencing in Figure 5b, that *nos* mRNA pool with short poly(A) tail increases in *aub* mutant due to lack of deadenylation, as we reported previously. We have clarified this point in the Results section p. 10.

We also show that Aub has an opposite role in polyadenylation on the pool of *nos* mRNA localized in the germ plasm (with longer poly(A)). We believe that Wisp might be in the Aub complex in the soma, and the new Wisp/Aub co-IP in *tud* mutant embryos are consistent with this. However, Wisp activity would be repressed in the soma, compared to in the germ plasm, by other components of the somatic complex (e.g. Smg). We have added this information in the Discussion p. 12.

According to the respective *nos* poly(A) tail phenotypes of *aub* mutant (lack of deadenylation/stabilization: \approx 100,000 reads (Figure 5b)) and *wisp* mutant (strong shortening of poly(A) tails/decay: \approx 100 reads (Supplementary Figure 5b)), there should be other RNA binding protein(s) recruiting Wisp to mRNAs in the soma.

9) Is the Aub-wispy interaction required for polyadenylation marks seen in wildtype embryos?

As indicated in point 8, the respective poly(A) tail phenotypes of *aub* and *wisp* mutant indicate that other proteins should interact with Wisp for polyadenylation in the soma. Regarding Wisp-dependent polyadenylation in the germ plasm, our additional experiments showing that the level of Wisp correlates with that of Aub in the germ plasm (Supplementary Figure 4c), and that posteriorly localized Aub is important for polyadenylation (Figure 5c and Supplementary 5c) strengthen the role of Aub/Wisp interaction for polyadenylation in the germ plasm.

Minor points

1. While it is important and nice to see that data by others can be recapitulated, the amount of new results displayed in the major figures in this manuscript is less than 1/2 of total data in the manuscript. Many results shown in this manuscript are recapitulation of previous published data some even in the main figures. i.e.: Fig 1 and most of Fig 2 was already reported previously (Macdonald, Simonelig and others) The fact that *csul* mutants affect *nos* RNA localization is already known (Mechler). The effect of

Tud on aub localization had already been demonstrated (Mourelatos, Lehmann, Xu).

The new data presented in this manuscript build on a number of previously published data. The careful quantification of these data and their quantitative comparison with our new experiments were mandatory to interpret and conclude on our new results, which led to identify the novel role of piRNAs in mRNA stabilization.

2. Handler et al., 2011 (EMBO J) demonstrated and corrected previous results and showed that Tud is not part of the piRNA pathway as the authors state in their introduction.

We have made this correction in the Introduction p. 3.

Together, the reported findings and the conclusions seem somewhat preliminary. It would be great if the authors could provide more direct evidence to distinguish between a role of Aubergine in localization versus protection of stability or both.

The idea that germ plasm tethered polyadenylation and protection from degradation contributes to the enrichment of localized RNAs is attractive and adds to similar conclusions made by Lipshitz for the *hsp83* RNA. Lipshitz and colleagues showed that *hsp83* RNA is enriched in the germ plasm because of lack of degradation but independent of localization. Indeed, this RNA may be a good target to study the relative role of Aub enrichment due to lack of degradation separate from localization. Some additional data along these lines could substantially extend the conclusion drawn in the manuscript.

As indicated in Major points 1 and 2, we have now discussed the role of Aub in the different localization mechanisms (Discussion p. 12).

We did sequence *Hsp83* mRNA poly(A) tails in *ob/+* and *ob/+; aub-* and wild-type embryos. These data showed that for *Hsp83* there is no specific pool of mRNA with longer poly(A) tails, therefore we could not specifically follow the pool of posteriorly localized/stabilized mRNA. Nonetheless, deadenylation was reduced in *ob* embryos compared to wild type at 1-2 h, which was consistent with a role of polyadenylation to counterbalance deadenylation in the germ plasm.

We also performed *Hsp83* RNA in situ in *ob/+* and *ob/+; aub-* embryos. Defective deadenylation in the soma in *aub* mutant embryos did not allow to properly quantify enrichment defects in the germ plasm. We did not include these data.

Reviewer #2

The manuscript by Dufourt et al. reports a new mechanism of Aub/piRNA-dependent mRNA stabilization in the germ plasm of the *Drosophila* embryo. This mechanism is opposite to the known Aub-dependent destabilization of mRNAs in the somatic part of the embryo. The authors showed that Aub achieves the stabilization function in the germ plasm by directly interacting with the Wisp poly(A) polymerase that elongates poly(A) tail and stabilizes target mRNAs. The experiments are well designed and the results are overall convincing. The findings reported in this manuscript are exciting, with general significance to developmental biology and RNA biology. I support the publication of the manuscript in Nature Communication without major revision.

Before the manuscript is accepted, I suggest a minor revision to address the following points:

(1) P 4, ll 2-3: "Aub-dependent unstable mRNAs encode germ cell determinants that undergo selective stabilization." This sentence should be revised because it is the mRNAs, but not germ cell determinants, undergo selective stabilization.

We have included this correction p. 4.

(2) Fig 1d: Show the percentage of Osk and nos mRNA in WT embryos with respect to normal, faint and no localization.

We have added this control in Figure 1d.

(3) P 6, para 2: "Taken together, these results show that Aub and piRNAs play a direct role in the localization of germ cell mRNAs in the germ plasm in late oocytes and/or early embryos." This is overstated. Up to now, the authors have only shown the role of Aub, but not piRNA, in the germ granule localization.

We have changed this sentence p. 6. We kept the information on piRNAs though in lines with the data

that we show with *armi* in this section.

(4) P 6, para 3, ll 9-11: “These results argue that Aub binding to mRNAs depends on sequence-specific targeting by piRNAs, rather than on random targeting along the entire length of mRNAs through very low complementarity, as was proposed recently”. This is over-concluded as well, since the localization of nos mRNA was reduced in only 36%- 37% of embryos deleted for both piRNA target sites. Hence, it still remains possible that some other piRNA sites, possibly in a semi-random fashion, are also required for the localization. Relevantly, the Aub-iCLIP data in Suppl Fig 2C are not very robust.

We have modified and clarified this part p. 7.

(5) P 9, Discussion: The role of piRNAs in cellular mRNA and lncRNA regulation in mammalian systems (ref. 16) should be mentioned here along with silkworm and Drosophila studies to provide a more intellectually complete picture.

We have included the information on the role of piRNAs in mouse sperm p. 11.

Reviewer #3

Although the conventional wisdom for the regulatory function of the Piwi pathway is transposon and gene silencing, a body of literature also implicates the Piwi pathway in an RNA localization role for germ plasm RNAs. Since some Argonaute proteins in nematodes have genetic roles in promoting gene expression, the notion that Piwi proteins could stabilize or promote gene expression is a very intriguing possibility that has not yet been fully described in other animals. The Simonelig lab has carried out previous studies of Aubergine-targeted germ plasm RNAs that are subjected to deadenylation during early embryogenesis, which follows the conventional gene silencing/destabilization role for Argonaute/Piwi proteins.

In this study, the authors now examine a potential RNA stabilization role for Aubergine in the germ plasm that is located at the posterior pole of the oocyte/embryo. Whereas Osk protein serves as an Aub-independent posterior germ plasm localization control, Nanos (nos) mRNA localization at the posterior pole is disrupted in Aub and *Armi* mutants. Interestingly, although *mnk* double mutants can partially rescue Osk posterior localization, no rescue is observed for nos mRNA.

Since Aub localization has been previously connected to arginine methylation by *Csul* and *Tud*, these mutations are examined for Aub protein and nos mRNA localization. The data in Fig 3 is very nice but restricted to just *Csul*, so I found it a bit odd that the *Tud* data is relegated to Sup Fig 3. A suggestion that seems logical to me is that the two mutants data could go together a single Figure 3, even though it would be a complicated figure. However, I raise one major issue here:

As requested, we have added the *tud* data in Figure 3.

Issue#1: On Pg 7, the authors propose the hypothesis that localized Aub at germ plasm is methylated while unmethylated Aub is in soma. But the only experiment assessing Aub arginine dimethylation is showing in the *csul* mutant decreased posterior accumulation of Aub and nos mRNA and decreased nos mRNA deadenylation. However, Aub is also destabilized in the *csul* mutant, according to Kirino et al, 2009. The analysis of Tudor mutants however do not directly address arginine demethylation state even if it does affect Aub and nos mRNA localization but not nos mRNA deadenylation. What about embryo staining and western blots with the SYM11 antibody from *csul* and *tud* mutants to show that arginine dimethylation is unaffected while Aub and nos mRNA posterior localization is greatly affected? This would be the key experiment to allow the authors to speak to the hypothesis they had proposed.

We have performed embryo immunostaining with SYM11 antibody, which show that indeed arginine dimethylation is strongly reduced in *csul* mutants, but not in *tud* mutant embryos (Supplementary Figure 3c).

Western blots with SYM11 did not allow to specifically identify Aub (using *aub* mutant embryos as controls). Indeed, although a western blot in Kirino et al. 2009 (Nature Cell Biol. 11, 652) suggested visualization of Aub with SYM11 antibody, data from Nishida et al. 2009 (EMBO J. 28, 3820) showed that this band was not Aub (Figure 2C).

In the last set of Figures 4 and S4, the authors test the interaction of Wispy, an embryonic PolyA

polymerase, and demonstrate compelling Aub interactions. Finally, the authors conduct an mPAT assay that uses deep-sequencing to measure PolyA tail length for specific transcripts, and this is conducted in Aub and wispy mutant embryos. Here, I found three significant issues to raise:
Issue#2: The data of Fig 4a-d, the ColP and colocalization of wispy with Aub very intriguing, but the study is still missing a test for whether these stainings and colPs are disrupted in the tud or csul mutant backgrounds.

Thank you for this suggestion. We have added these data in Figure 4e and Supplementary Figure 4c. Both data provide important informations that strengthen the manuscript. They are discussed on p. 9 in the Results section, and on p. 12 in the Discussion.

Issue#3: The mPAT assay cumulative distribution profiles need to have highlighting of which regions are significantly different between the time points and mutants vs genetic controls. The authors need to discuss and reconcile the inconsistency in the mPAT profiles with the traditional gel-based PAT assay, comparing Fig 4f and 4g. In particular, there is a concerning saturation of reads in the mPAT assay shorter than 50 PolyA's when there is no such saturation in the traditional PAT assay, and there are incongruency in the profiles. Marking the meaningful changes in distribution plots in Fig S4 is also needed.

We have indicated the regions of interest with long poly(A) tails in the graphs of mPAT in Figure 5 and Supplementary Figure 5. The saturation of short reads in the mPAT is because the graphs are represented as cumulative, meaning that the read counts with 4A (the shorter poly(A) tail taken into account) represent all the reads, then read counts with 5A correspond to all the reads that have at least 5A, etc... We have better explained this representation in the legend of Figure 5b, c, and we have added a scheme in Figure 5b, c.

Issue #4: The authors should conduct and describe an analysis on how much overlap in Aub iCLIP targets and mRNA targets deregulated in PolyA tails in the Wispy mutant, from Eichorn et al 2016 study? This should provide better scope as to how many of the overall Wispy targets are also subjected to Aub regulation.

We have performed this comparison that is shown in Supplementary Figure 5d and reported in the Discussion p. 12.

Overall, I enjoyed reading this paper and found most of the data and story very interesting. I recommend the authors address all the main issues I raised in order to satisfy my full support of this study.

REVIEWERS' COMMENTS:

Reviewer #2 (Remarks to the Author):

The revision has satisfactorily addressed my comments. I support its publication.

Reviewer #3 (Remarks to the Author):

I am satisfied with most of the author's responses except for the response to the Issue #3 that I raised, in that the mPat plots in now Fig 5 and Supp Fig 5 which the author's claim are cumulative distribution profiles are NOT true Cumulative Distribution Function (CDF) plots (I did not notice earlier that the Y-scale is a log-scale of just read counts, not a true fraction scale). Lim et al 2016 Fig 1 and Fig2 as well as Eichorn et al 2016 Fig 4 has some key examples of true CDF plots, which show the major peak in the CDF at ~50nt that would be entirely consistent with the traditional gel-based PAT assay. However, this other plotting method in the original manuscript and this revision is not a correct CDF plot, and the diagram schematic is NOT helpful at supporting or illustrating the plot, hence my original impression there was some incongruency in the mPAT profiles versus the gel-base PAT assay.

This is because the analysis is not quite correct although able to show indications in Poly-A tail length changes. The authors are incorrect to call these CDF plots (and I apologize for not catching the actual root of the error earlier, but still not too late). If the authors are unable to redo just the analysis of the data into true CDF plots, then they should change the plot descriptions to some other plot and discuss and reconcile the inconsistency in these mPAT profiles with the traditional gel-based PAT assay rather than just a readjustment of the figures.

Reviewer #4 (Remarks to the Author):

In their manuscript "piRNAs and Aubergine cooperate with the Wispy poly(A) polymerase to stabilize mRNAs in the germ plasm", Simonelig and colleagues report on a role for Aub and Wispy (poly-A polymerase) in mRNA stabilization in germ plasm. This role in mRNA protection extends roles of Piwi family proteins outside of transposon silencing, and also is unconventional in the sense that Argonaute proteins in general are usually known for functions in transcript destabilization.

The topic is interesting and appropriate for Nature Communications. The previous referees have raised many relevant issues regarding this work, and the authors have made substantial responses and revisions. Although some of the findings have been described previously, it is useful to conduct them as a foundation and so that is fine as long as citations are provided as raised by the referees.

Overall the potential for a mechanism involving Aub to recruit Wispy to polyadenylate targets is intriguing and maybe the most important part. However, since their inputs show many bands besides their intended full-length and they did not purify the proteins, I don't think its appropriate to show some of the very tightly cropped boxes in the figure. They should show more of the blots/inputs and label the bands of interest in the main figure as appropriate and not always "hide" the other bands. Otherwise most readers are not going to dig into the supplementary data and appreciate the experiment is not as clean.

In general, the topic of how mRNA targeting in soma and germ plasm by Aub can have differential functional outcomes is intriguing but difficult to disentangle due to limited and imprecise genetic reporters. As mentioned by the referees and in author discussion, this study does not resolve the Aub-trap model, nor is it clear exactly what effect $\text{nos}(\Delta\text{piroo}\Delta\text{pi412})$ has on Aub binding. The

authors present an interesting advance on their prior model, and in future studies they will need better and more precise genetic reporters for functional tests.

Reviewer #2

The revision has satisfactorily addressed my comments. I support its publication.

Reviewer #3

I am satisfied with most of the author's responses except for the response to the Issue #3 that I raised, in that the mPat plots in now Fig 5 and Supp Fig 5 which the author's claim are cumulative distribution profiles are NOT true Cumulative Distribution Function (CDF) plots (I did not notice earlier that the Y-scale is a log-scale of just read counts, not a true fraction scale). Lim et al 2016 Fig 1 and Fig2 as well as Eichorn et al 2016 Fig 4 has some key examples of true CDF plots, which show the major peak in the CDF at ~50nt that would be entirely consistent with the traditional gel-based PAT assay. However, this other plotting method in the original manuscript and this revision is not a correct CDF plot, and the diagram schematic is NOT helpful at supporting or illustrating the plot, hence my original impression there was some incongruency in the mPAT profiles versus the gel-base PAT assay.

This is because the analysis is not quite correct although able to show indications in Poly-A tail length changes. The authors are incorrect to call these CDF plots (and I apologize for not catching the actual root of the error earlier, but still not too late). If the authors are unable to redo just the analysis of the data into true CDF plots, then they should change the plot descriptions to some other plot and discuss and reconcile the inconsistency in these mPAT profiles with the traditional gel-based PAT assay rather than just a readjustment of the figures.

We agree with the reviewer that the mPAT plots shown in Figures 5 and S5 are not Cumulative Distribution Function (CDF) plots because they are not scaled to 100%. We have therefore changed the legend of Figures 5 and S5 to indicate that these plots are "cumulative plots of sequenced poly(A) tails". mPAT profiles are consistent with the ePAT gel shown in Figure 5a. Both data show increased levels of short poly(A) tails (<50) and decreased levels of long poly(A) tails (>80) in *aub* mutant as compared to wild-type embryos.

Reviewer #4

In their manuscript "piRNAs and Aubergine cooperate with the Wispy poly(A) polymerase to stabilize mRNAs in the germ plasm", Simonelig and colleagues report on a role for Aub and Wispy (poly-A polymerase) in mRNA stabilization in germ plasm. This role in mRNA protection extends roles of Piwi family proteins outside of transposon silencing, and also is unconventional in the sense that Argonaute proteins in general are usually known for functions in transcript destabilization.

The topic is interesting and appropriate for Nature Communications. The previous referees have raised many relevant issues regarding this work, and the authors have made substantial responses and revisions. Although

some of the findings have been described previously, it is useful to conduct them as a foundation and so that is fine as long as citations are provided as raised by the referees.

Overall the potential for a mechanism involving Aub to recruit Wispy to polyadenylate targets is intriguing and maybe the most important part. However, since their inputs show many bands besides their intended full-length and they did not purify the proteins, I don't think its appropriate to show some of the very tightly cropped boxes in the figure. They should show more of the blots/inputs and label the bands of interest in the main figure as appropriate and not always "hide" the other bands. Otherwise most readers are not going to dig into the supplementary data and appreciate the experiment is not as clean.

In general, the topic of how mRNA targeting in soma and germ plasm by Aub can have differential functional outcomes is intriguing but difficult to disentangle due to limited and imprecise genetic reporters. As mentioned by the referees and in author discussion, this study does not resolve the Aub-trap model, nor is it clear exactly what effect $nos(\Delta\pi roo\Delta\pi 412)$ has on Aub binding. The authors present an interesting advance on their prior model, and in future studies they will need better and more precise genetic reporters for functional tests.

As requested by the reviewer, we have included the complete gels for input proteins in Figure 4f and labelled the bands of interest. We have also toned down the Discussion on p. 11, 12 and 13.